# Apolipoprotein C3-Rich Low-Density Lipoprotein Induces Endothelial Cell Senescence via FBXO31 and Its Inhibition by Sesamol In Vitro and In Vivo

**DOI:** 10.3390/biomedicines10040854

**Published:** 2022-04-05

**Authors:** Ping-Hsuan Tsai, Li-Zhen Chen, Kuo-Feng Tseng, Fang-Yu Chen, Ming-Yi Shen

**Affiliations:** 1Graduate Institute of Biomedical Sciences, China Medical University, Taichung 40402, Taiwan; u105010312@cmu.edu.tw (P.-H.T.); throsten.trista@gmail.com (L.-Z.C.); fyc0321@gmail.com (F.-Y.C.); 2Department of Biological Science and Technology, China Medical University, Taichung 40402, Taiwan; u107010409@cmu.edu.tw; 3Department of Medical Research, China Medical University Hospital, Taichung 40402, Taiwan; 4Department of Nursing, Asia University, Taichung 41354, Taiwan

**Keywords:** apolipoprotein C3-rich LDL, atherosclerosis, FBXO31, premature endothelial senescence, sesamol

## Abstract

Premature endothelial senescence decreases the atheroprotective capacity of the arterial endothelium. Apolipoprotein C3 (ApoC3) delays the catabolism of triglyceride-rich particles and plays a critical role in atherosclerosis progression. FBXO31 is required for the intracellular response to DNA damage, which is a significant cause of cellular senescence. Sesamol is a natural antioxidant with cardiovascular-protective properties. In this study, we aimed to examine the effects of ApoC3-rich low-density lipoprotein (AC3RL) mediated via FBXO31 on endothelial cell (EC) senescence and its inhibition by sesamol. AC3RL and ApoC3-free low-density lipoproteins (LDL) (AC3(-)L) were isolated from the plasma LDL of patients with ischemic stroke. Human aortic endothelial cells (HAECs) treated with AC3RL induced EC senescence in a dose-dependent manner. AC3RL induced HAEC senescence via DNA damage. However, silencing FBXO31 attenuated AC3RL-induced DNA damage and reduced cellular senescence. Thus, FBXO31 may be a novel therapeutic target for endothelial senescence-related cardiovascular diseases. Moreover, the aortic arch of hamsters fed a high-fat diet with sesamol showed a substantial reduction in their atherosclerotic lesion size. In addition to confirming the role of AC3RL in aging and atherosclerosis, we also identified AC3RL as a potential therapeutic target that can be used to combat atherosclerosis and the onset of cardiovascular disease in humans.

## 1. Introduction

Age is a dominant risk factor for atherosclerotic cardiovascular disease (CVD), coronary heart disease, and stroke [1,2]. Endothelial cell (EC) senescence is closely associated with atherosclerosis and is an adverse outcome in coronary heart disease and stroke [3,4,5]. Human somatic cells in culture have a finite replicative lifespan, which was the original definition of cellular senescence. Senescent cells enter permanent growth arrest and exhibit a flattened and enlarged morphology. In senescent cells, p53 and p21 (negative regulators of the cell cycle) expressions are increased [2]. Senescent cells exhibit phenotypic changes not seen in quiescent cells, which have been linked to age-related vascular diseases, such as atherosclerosis [6]. Vascular cell senescence appears to play an important role in atherogenesis, as evidenced in recent studies [2,6,7], even though atherosclerosis is considered to be a chronic inflammatory disease [8]. p53 modulates the expression of genes that regulate cell cycle progression, cell death, and survival [9]. Moreover, this pathway plays a critical role in human cell senescence. Mouse double minute 2 homolog (MDM2; also known as E3 ubiquitin–protein ligase) is an oncoprotein that serves as a major negative regulator of p53 [10]. MDM2 inhibits p53 cell cycle arrest and apoptosis. Through the accelerated proteasome-dependent degradation of p53, MDM2 can lead to a substantial decrease in p53 protein levels [11]. The FBXO31-mediated loss of MDM2 leads to elevated p53 levels, resulting in growth arrest [12]. It is likely that cellular senescence processes in atherosclerosis are multiple and additive in each cell and artery that they affect [4].

Sesamol (SM), a crucial active ingredient in sesame seed oil, is a dietary compound that is soluble in both the aqueous and lipid phases [13]. Sesamol has been identified as a healthy food and is used as a traditional medicine. It has previously been shown to have antimutagenic, anticancer, anti-atherosclerotic, hepatoprotective, and antiaging properties mediated by reactive oxygen species. In other studies, it was found to inhibit lipid peroxidation, hydroxyl radical-induced deoxyribose degradation, and DNA cleavage [13].

Apolipoprotein C3 (ApoC3) acts as an independent risk factor for coronary heart disease, particularly when it is present as a component of ApoB lipoprotein [14]. Excessive ApoC3 can postpone the lipolysis of ApoB lipoprotein [15] and inhibit its uptake by the normal high-affinity receptors of ApoB lipoprotein on hepatocytes, causing hypertriglyceridemia [16]. In addition, ApoC3 containing LDL prepared from human plasma activates monocytes that circulate in the blood to adhere to vascular endothelial cells, an early step in atherosclerosis [17,18]. In the present study, we explored the role of ApoC3-rich LDL (AC3RL) in vascular cell senescence in human aortic endothelial cells (HAECs) and tested our hypothesis that AC3RL is a key player in early atherogenesis. Additionally, we examined the effects of AC3RL, mediated by FBXO31, on endothelial cell senescence and its inhibition by SM.

## 2. Materials and Methods

### 2.1. LDL Isolation

At the time of the collection of blood samples for the study, the blood samples were preserved in ethylenediamine tetra-acetic acid (EDTA), 1% ampicillin/streptomycin, and aprotinin (Sigma-Aldrich, St. Louis, MO, USA) to prevent contamination. Whole blood was centrifuged at 400× *g* for 15 min at 4 °C to obtain the plasma. Sequential potassium bromide density centrifugation was used to separate LDL from other lipids, such as chylomicrons, very low-density lipoproteins (VLDL), and intermediate-density lipoprotein (IDL) fractions [19]. LDL was treated with nitrogen and EDTA (5 mM) to prevent ex vivo lipid oxidation.

### 2.2. Isolation and Measurement of AC3RL

Polyclonal goat anti-human ApoC3 antibodies (33A-G2b, Academy BioMedical, Houston, TX, USA) coupled to the Sepharose 4 B resin (Academy Biomedical) were incubated overnight with LDL. Gravity flow was used to collect lipoproteins without ApoC3, and 3 mol/L sodium thiocyanate (Sigma-Aldrich) was used to elute lipoproteins with ApoC3 [20]. ApoC3 immunoaffinity separation worked well with a 99% success rate. ELISA was used to assess the levels of ApoB (SC-13538, Santa Cruz Biotechnology, Santa Cruz, CA, USA) and ApoC3 in LDL-containing and LDL-excluding ApoC3.

### 2.3. Cell Culture and Treatment

EGM2 medium (Promo Cell, Heidelberg, Germany) was used to culture primary HAECs for cell studies [21]. To perform any cellular experiment, the HAECs had to be grown in 12-well plates until they reached 80% confluency. To evaluate the effect of AC3RL on cellular senescence, HAECs were incubated with sub-apoptotic levels of AC3RL (20 μg/mL), the same concentration of AC3(-)L, and PBS (lipoprotein-free control) for two consecutive days. Cells treated with AC3RL also received a dose of SM (0.3–3 μM) to assess anti-senescence effects.

### 2.4. SA-β-Gal Staining for HAECs

SA-β-Gal staining was performed using a senescence-associated (SA) β-galactosidase staining kit (Cell Signaling Technology, Beverly, MA, USA) according to the manufacturer’s instructions [22]. The percentage of SA-β-Gal-positive cells was determined by counting over 30,000 cells in nine randomly selected fields under a microscope (Olympus, Tokyo, Japan). Senescent cells were identified as the blue-stained cells.

### 2.5. Western Blotting Analysis

For all Western blotting analyses, HAECs were collected and lysed in a radioimmunoprecipitation assay (RIPA) buffer (Pierce, Rockford, IL, USA) containing a protease inhibitor cocktail. The protein concentrations were determined using a BCA assay (Pierce). Samples were separated by SDS-PAGE (4–20% gels; Invitrogen, Carlsbad, CA, USA) and transferred to a PVDF membrane. Primary antibodies, including anti-FBXO31 (A302-047A, Bethyl Laboratories, Montgomery, TX, USA, 1:1000), anti-p-MDM2 (#3521, Cell Signaling Technology, 1:1000), anti-p53 (#2524, Cell Signaling Technology, 1:1000), anti–p21 (GTX62525, GeneTex, Alton Pkwy Irvine, CA, USA, 1:1000), and anti-β-actin (A5441, Sigma-Aldrich, 1:5000), were used. All signals were detected using an ECL reagent (Millipore, Bedford, MA, USA). Statistical results were obtained by scanning the reactive bands and measuring the density of optical cells using video densitometry (G-box Image System, Syngene, Frederick, MD, USA) [23].

### 2.6. Immunofluorescence Microscopy

HAECs or hamster aortic tissue sections were fixed in 4% paraformaldehyde, permeabilized with 0.2% Triton X-100, and blocked with SuperBlock (Pierce). Immunostaining was performed by incubating them with anti-γH2AX monoclonal antibodies (Cell Signaling Technology) for 2 h at 4 °C, followed by incubation with Alexa-488 conjugated secondary antibodies for 1 h at 37 °C. Hoechst 33342 was used to stain the nucleic acids. Anti-MDM2 antibodies (SC-965, Santa Cruz Biotechnology) were used to detect in situ MDM2 in aortic tissue samples. γH2AX foci were counted visually in >50 cells by capturing images of randomly chosen fields, and cells with ≥3 foci were defined as positive for γH2AX foci [24]. Images were captured using an IX70 inverted microscope (Olympus, Tokyo, Japan). Each set of images was captured using the same set of imaging settings. ImageJ software was used to measure the mean fluorescence intensity.

### 2.7. Animals, Groupings, SA-β-Gal Staining, and Oil Red O Staining

Syrian hamsters (NARLabs, Taipei, Taiwan) that had LDL characteristics comparable to those of humans were fed a normal chow diet (control), high-fat diet (HFD), or HFD supplemented with 50 or 100 mg/kg SM via oral gavage (HFD + SM) for 18 weeks (*n* = 10 per group). Among these groups, we compared aortic endothelial senescence in the descending aorta using SA-β-Gal staining and the atherosclerotic lesion size in the aortic arch using Oil Red O staining.

### 2.8. Immunoprecipitation

ApoC3 antibodies (ab7619, Abcam, Cambridge, UK) were added to 50 μL of magnetic Dynabeads (Qbeads Protein G; MagQu, Taipei, Taiwan) and incubated overnight at 4 °C on a rocking platform. The plasma was treated with antibody-coated beads and the beads were pelleted and washed to remove any remaining antibodies. ApoC3 was isolated and pelleted using a magnetic particle concentrator after rinsing with elution buffer (20 μL, 0.1 M Glycine-HCl, pH 2.0). The ApoC3-containing supernatant was collected. A neutralization buffer (2 μL, 1 M Tris-HCl, pH 8.5) was added to adjust the pH of the elute [19].

### 2.9. mRNA Analysis Using Real-Time Quantitative PCR

Following the homogenization of the hamster aorta, samples were frozen in NucleoZOL at −80 °C, and total RNA was isolated and extracted according to the manufacturer’s instructions [25]. cDNA synthesis was performed using the iScript cDNA Synthesis Kit (Bio-Rad, Hercules, CA, USA). Finally, real-time PCR was performed using Q SYBR Green Supermix (Bio-Rad). A list of primers used in real-time PCR tests is given in Table 1.

### 2.10. Data Analysis and Statistical Procedures

Student’s *t*-test was used to determine the statistical significance of the differences between the groups. Probability values of *p* < 0.05 were regarded as statistically significant in this study. The results are expressed as mean ± standard deviation.

## 3. Results

### 3.1. AC3RL Induces Endothelial Cell Senescence

To examine the effect of AC3RL on endothelial cell senescence, we used AC3(-)L or AC3RL (25 μg/mL) isolated from the plasma of stroke patients to treat HAECs for 24 h. In AC3RL-treated cells, blue deposits after senescence-associated (SA)-β-Gal staining indicated that AC3RL, but not AC3(-)L, significantly induced endothelial cell senescence (Figure 1A, *p* < 0.001). In addition, the expression of senescence molecular markers, p53 and p21, in endothelial cells was elevated after treatment with AC3RL (Figure 1B,C). However, pretreatment with 5 mM n-acetyl cysteine (NAC, a strong antioxidant), attenuated AC3RL-induced cell senescence (Figure 1B,C). This finding suggested that oxidative stress may be involved in AC3RL-induced endothelial cell senescence.

### 3.2. ROS Is Involved in AC3RL-Induced EC Senescence

To examine whether oxidative stress was involved in AC3RL-induced senescence, we used DCFH-DA staining to evaluate the effect of AC3RL on intracellular oxidative stress. DCFH-DA staining showed that AC3RL, but not AC3(-)L, significantly increased intracellular oxidative stress (Figure 2A, *p* < 0.001), whereas NAC or SM reduced AC3RL-increased intracellular ROS production (Figure 2A, *p* < 0.001). Because oxidative stress is a strong trigger for the DNA damage response and subsequent cellular senescence, we used immunofluorescence to examine the expression of γH2AX in HAECs after AC3RL treatment. Immunofluorescence staining for γH2AX showed that AC3RL, but not AC3(-)L, significantly increased the intracellular oxidative stress levels (Figure 2B,C, *p* < 0.001), whereas NAC or SM inhibited AC3RL-increased intracellular oxidative stress in HAECs (Figure 2B,C, *p* < 0.001). Therefore, our results indicate that AC3RL induced EC senescence through oxidative stress and subsequent DNA damage, whereas SM might inhibit these effects. 

### 3.3. The Role of FBXO31 in AC3RL-Induced Cell Senescence

FBXO31 plays a crucial role in the regulation of DNA damage response-related proteins, such as MDM2 and p53. Thus, to evaluate the role of FBXO31 in AC3RL-induced HAEC senescence, we used siRNA to knockdown FBXO31 expression. SA-β-gal staining demonstrated that the knockdown of FBXO31 expression arrested AC3RL-induced EC senescence (Figure 3A). Moreover, treatment with siFBXO31 reduced AC3RL-induced oxidative stress (Figure 3B) and DNA damage (Figure 3C). Regarding the molecular mechanism, the inhibition of the expression of FBXO31 by siFBXO31 not only reduced the AC3RL-induced phosphorylation of MDM2, but it also decreased the expression of p53 and p21 (Figure 4A,B). Moreover, NAC (5 mM) or SM (3 μM) also decreased the AC3RL-induced expression of FBXO31 and the phosphorylation of MDM2 in HAECs (Figure 4C). Our results indicate that the genetic manipulation of FBXO31 arrested AC3RL-induced HAEC senescence. SM also prevented AC3RL-induced HAEC senescence via FBXO31 inhibition.

### 3.4. AC3RL-Induced Vascular Endothelial Senescence and the Inhibitory Effect of Sesamol

To confirm the effects of AC3RL-induced senescence in vivo, we used a high-fat diet (HFD) to induce dyslipidemia in Syrian hamsters. The Syrian hamsters were fed a chow diet (control), an HFD, or an HFD supplemented with 50 or 100 mg/kg SM via oral gavage (HFD + SM) for 18 weeks (*n* = 10 per group). First, we analyzed the apolipoprotein component of LDL in each treatment hamster. We found that the hamsters of the HFD group had a higher ApoC3 content in their LDL than that of the control group (Figure 5A). These data indicate that the hamster HFD group had higher levels of endogenous AC3RL than the control group. Moreover, we examined aortic endothelial senescence and the atherogenic properties of AC3RL. SA-β-gal staining and Oil Red O staining of the thoracic aorta or arch aorta of hamsters showed endothelial senescence and lipid accumulation in the HFD group (Figure 5B,C). Next, we examined DNA damage, FBXO31, and the MDM2 expression of AC3RL in vascular tissue. Immunofluorescence staining showed nuclear γH2AX deposition and MDM2 protein downregulation on the luminal side of the thoracic aorta and the *MDM2* mRNA down-expression and *FBXO31* mRNA up-expression of thoracic aorta tissue in the HFD group. However, oral gavage of SM (50 mg/kg or 100 mg/kg) administered to the hamster HFD group remarkably reduced ApoC3 expression among the LDL (Figure 5A); arrested vascular endothelial senescence (Figure 5B), lipid accumulation (Figure 5C), and protein expression (Figure 5D); and reduced *FBXO31*, *p53*, and *p16* mRNA expression and reversed *MDM2* mRNA expression (Figure 5E–H). These hamsters also showed decreased nuclear γH2AX deposition and MDM2 expression, compared to those in the hamster HFD group, in the immunofluorescence staining (Figure 6A,B). 

## 4. Discussion

In this study, we demonstrated, for the first time, that AC3RL induces EC senescence both in vivo and in vitro. In addition, we showed that HFD led to ApoC3 overexpression on LDL (elevated plasma level of AC3RL) and aortic endothelial senescence in hamsters. These findings indicate that AC3RL may play a causal role in endothelial senescence and atherosclerosis formation. Moreover, we showed that in cultured HAECs, AC3RL-induced endothelial senescence was mediated by intracellular ROS formation, γH2AX deposition, and FBXO31 activation, resulting in the inhibition of MDM2, p53, and p21 activation. However, SM may reduce ROS levels and inhibit FBXO31 activation to arrest AC3RL-induced endothelial senescence and atherosclerotic formation (Figure 7).

Coronary artery endothelial cells from plaques in patients with ischemic heart disease were found to be senescent [6,7]. Previous studies have found that vascular senescence may contribute to the remodeling of adverse vasculature and stiffness before the development of atherosclerosis [4]. AC3RL is an LDL-containing ApoC3. A previous study reported that ApoC3 delays the catabolism of triglyceride-rich particles, which is crucial for the advancement of atherosclerosis [17]. In this study, we found that AC3RL induced cellular senescence in HAECs. In addition, we found that p53 and p21 levels in endothelial cells were elevated after treatment with AC3RL. The expression of p53 and p21 are considered to be molecular markers of cell senescence [26]. Moreover, p53 is a protein with short-lived properties that is maintained at low levels in normal cells. DNA damage responds to a rapid increase in p53 levels and the subsequent inhibition of cell growth [27]. p21 is a primary target of p53; thus, it is correlated with cell cycle arrest and senescence as a result of DNA damage [26]. In addition, vascular aging and atherosclerosis are exacerbated by ROS-induced oxidative stress and the DNA damage response [4,28]. We indicated that AC3RL increased intracellular oxidative stress by using DCFH-DA staining and γH2AX overexpression in HAECs. These data imply that AC3RL-mediated vascular or endothelial senescence may be due to oxidative stress and endothelial DNA damage.

FBXO31 is a subunit of the RING finger SCF4 (SKP1-Cullin 1-F-box protein) E3 ubiquitin ligases complex [29] and is involved in the regulation of DNA damage response-related proteins [30]. The FBXO31-mediated loss of MDM2 leads to elevated p53 levels, resulting in growth arrest [12]. Additionally, the phosphorylation of MDM2 promotes its destruction by SCF^β-TRCP^ and leads to an increase in p53 protein levels [31,32]. In this study, the siFBXO31-mediated knocking down of FBXO31 could inhibit the phosphorylation of MDM2 and reduce the expression of p53. These cells also arrested AC3RL-induced endothelial oxidative stress, DNA damage, and cell senescence. These data indicate that AC3RL downregulates MDM2 in EC senescence. Furthermore, mechanisms that modulate the MDM2-induced degradation of p53 may help regulate the extent and duration of the p53 response. Moreover, these results suggested that the genetic manipulation of FBXO31 may arrest AC3RL-induced HAEC senescence.

Apolipoprotein C3 (ApoC3), a surface protein component abundantly found on circulating triglyceride-rich ApoB lipoproteins and HDL, is a major contributor to atherosclerosis, according to increasing evidence [17]. The plasma levels of ApoC3 and lipoproteins that carry ApoC3 independently predict a higher risk of coronary heart disease in prospective human cohorts after adjusting for blood lipids [14,33]. The overexpression of ApoC3 causes hyperlipidemia and promotes atherosclerotic lesion development in mouse models [34]. However, ApoC3 deficiency protects against dyslipidemia and atherogenesis [35]. Previous studies have reported that ApoC3 alone or as a component of VLDL or LDL induces monocyte activation and adhesion to ECs [17]. These observations suggest that ApoC3 may play a causative role in the development of atherosclerotic lesions independently of its detrimental effects on lipid metabolism. We examined the effects of AC3RL on human aortic EC senescence and its signaling pathway in vitro. To confirm the effects of AC3RL-induced senescence in vivo, we used an HFD to induce dyslipidemia in Syrian hamsters, which develop atherosclerosis in a manner that closely mimics human pathology [23]. In this study, we showed that the hamster HFD group produced higher levels of ApoC3 in their LDL than that in the control group. These data indicated that hamsters had higher levels of endogenous AC3RL after HFD feeding. However, the exact mechanism underlying the elevation of ApoC3 in the LDL in the hamster HFD group is not fully understood. From a previous study, we understand that abnormal lipid metabolism and insulin resistance affect ApoC3 secretion and recovery rates [36]. The liver is the main site of ApoC3 expression. ApoC3 expression in the liver is induced by carbohydrates (glucose and fructose) and saturated fatty acids (SFA). Carbohydrates and SFA not only increase ApoC3 expression, but also increase the hepatic de novo lipogenesis of fatty acids, which is a major cause of non-alcoholic fatty liver disease (NAFLD) and excess triglyceride-rich VLDL particle production. ApoC3 exists mostly in VLDL. VLDL is metabolized to LDL in the bloodstream. However, under normal metabolism, ApoC3 is rarely found in LDL [36]. Moreover, a high-fat diet raises plasma SFA levels, which are primarily responsible for pathological changes of abnormal lipid metabolism and insulin resistance [37]. Perhaps these are the mechanisms responsible for raising the ApoC3 in LDL in hamsters fed with an HFD. We also examined aortic endothelial senescence and the atherogenic properties of AC3RL. The results showed vascular endothelium senescence through thoracic aorta staining with SA-β-gal and lipid accumulation via arch aorta staining with Oil Red O in the HFD group. Moreover, we identified DNA damage (nuclear γH2AX deposition), while FBXO31 was increased and MDM2 was decreased in expression in the vascular tissue of the hamster with high-level endogenous AC3RL. The in vivo signaling of AC3RL-mediated cell senescence was similar to that in vitro. Our in vivo study indicated that dyslipidemia not only elevated the level of ApoC3 in the apolipoprotein component of LDL, but that it also induced vascular endothelial senescence and enlarged the atherosclerotic lesion size. These results are consistent with those of our in vitro studies, suggesting that AC3RL induces endothelial senescence and facilitates the progression of atherosclerosis.

Sesame is widely used in Chinese and Indian herbal medicines [23]. SM, which is isolated from sesame, reduces oxidative stress [38], has anti-hypertensive [39] and anti-hyperlipidemic potential [23], and is effective in preventing atherosclerosis and CVD [25]. In this study, we examined whether SM prevents EC senescence induced by ApoC3-rich LDL. We administered SM via oral gavage for 18 wks at physiological doses of 50 or 100 mg/kg body weight. Through the experiments, we showed that SM may markedly reduce the ApoC3 expression of LDL, endothelium senescence, and lipid accumulation; moreover, nuclear γH2AX deposition, FBXO31 reduction, and MDM2 reversal was identified in the vascular tissue of the hamster HFD with SM groups. Our previous studies demonstrated that it upregulates the phosphatidylinositol 3-kinase (PI3K)/Akt pathway [23]. The PI3K/Akt pathway may regulate the degradation of FBXO31 [40], promote the translocation of MDM2 from the cytoplasm to the nucleus, and decrease cellular levels of p53 [41]. In addition, our recent study also found that SM reduced p53 expression in HAECs [25]. Decreased p53 may inhibit cell senescence [31]. Collectively, our results reveal that SM inhibits AC3RL-induced EC senescence via FBXO31 reduction and the upregulation of the MDM2 pathway.

The accelerated development of atherosclerosis and premature aging of the cardiovascular system have been recognized in patients with atherosclerotic vascular disease and metabolic syndrome (MetS) [1,2,42]. Moreover, growing evidence suggests that cellular senescence is not only associated with atherosclerosis, but that it also promotes atherosclerosis [2,4,6]. It is likely that cellular senescence mechanisms in atherosclerosis are multiple and cumulative in each cell and artery that they affect. Atherosclerosis is linked to early biological aging, when atherosclerotic plaques exhibit signs of cellular senescence, such as diminished cell proliferation, irreversible growth arrest and apoptosis, increased DNA damage, epigenetic modifications, shortened telomeres, and malfunction. [4]. Targeting FBXO31 and the DNA damage response pathway may provide novel therapeutic opportunities for the prevention and treatment of atherosclerotic vascular diseases related to elevated plasma levels of AC3RL.

The absorption of plasma lipoproteins, as well as the expression of several pro-inflammatory molecules, is increased in aging vessels. In addition to atherogenic low-density lipoprotein, exposure to hypercholesterolemia or H_2_O_2_-induced ROS triggers premature senescence and atherosclerosis. Existing results from human clinical trials suggest that natural products can be used in conjunction with conventional therapies to reduce the concentration of small, dense LDL and LDL particles, thus helping to avoid CVD [43,44]. Owing to the general activation of oxidized LDL, inflammatory leukocytes (T cells and monocytes) are attracted to the subendothelial region, where they can bind and recruit cell adhesion molecules, including VCAM-1 and ICAM-1 [45]. Therefore, this platform can be used to examine the diseases and therapies associated with diabetes and dyslipidemia. SM protects against HAEC damage by lowering the effect of AC3RL-mediated ROS/FBXO31 on HAECs. Stress-inducing chemicals, such as reactive oxygen nitrogen species (RONS), hormones, and other medicines, may alter the protective action of SM. Alternatively, other mechanisms that may cause cell aging, including mitochondria-derived ROS or the repression of telomerase activity to induce senescence, must be investigated in detail. The various methods of SM action should also be explored in future research.

## 5. Conclusions

The critical findings of this study suggest that AC3RL induces endothelial senescence and facilitates atherosclerosis progression. SM, owing to its antioxidative effects, may provide protection against AC3RL-mediated atherosclerosis and the development of CVD in humans.

## Figures and Tables

**Figure 1 biomedicines-10-00854-f001:**
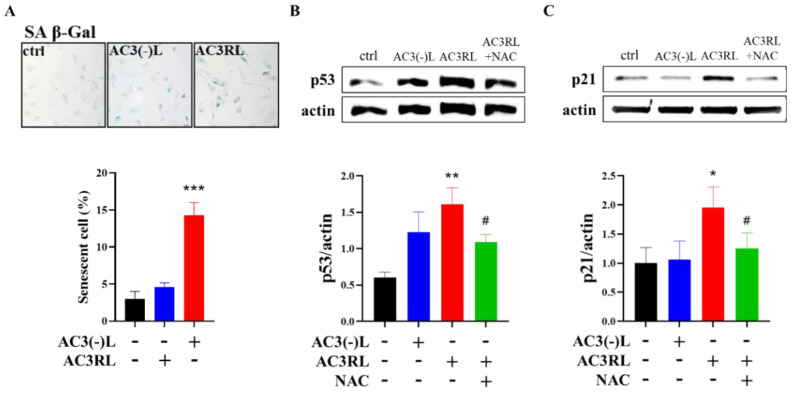
Effect of AC3RL on HAEC senescence. After treatment with AC3RL, AC3(-)L, or PBS, HAECs were stained with SA-β-Gal (blue) and positively stained cells were quantified in (**A**). A Western blotting assay was used to assess (**B**) p53 expression and (**C**) p21 expression. For the quantitative analyses, *n* = 3 per group. * *p* < 0.05, ** *p* < 0.01, *** *p* < 0.001 vs. control; # *p* < 0.05 vs. AC3RL.

**Figure 2 biomedicines-10-00854-f002:**
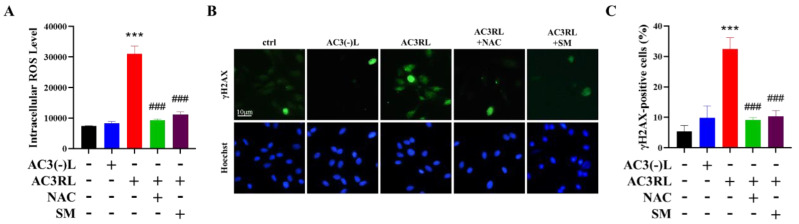
The reactive oxygen species (ROS) production and DNA damage effect of AC3RL in HAECs. HAECs were incubated with PBS (control, ctrl), AC3(-)L (25 μg/mL), AC3RL (25 μg/mL), and/or AC3RL + NAC (5 mM) or SM (sesamol, 3 μM) for 24 h. (**A**) Intracellular ROS production in HAECs. (**B**,**C**) Immunofluorescence staining showing γH2AX foci in HAECs from each treatment group. The fraction of positively stained cells is displayed as a percentage of the overall cell population. For the semi-quantitative analyses, *n* = 3 per group. *** *p* < 0.001 vs. control; ### *p* < 0.001 vs. AC3RL.

**Figure 3 biomedicines-10-00854-f003:**
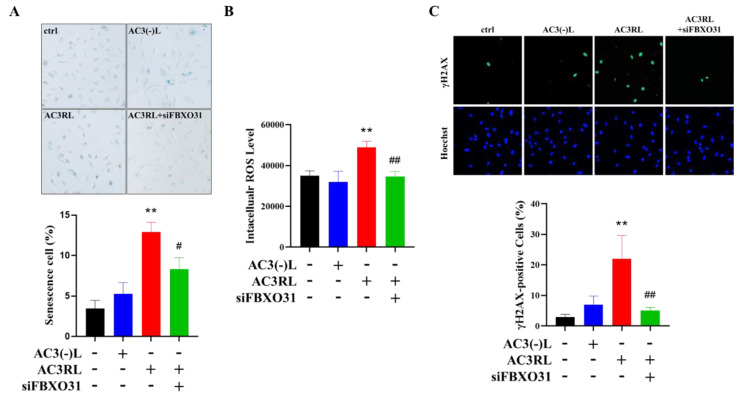
AC3RL induced cell senescence via FBXO31. HAECs were incubated with PBS (control, ctrl), AC3(-)L (25 μg/mL), AC3RL (25 μg/mL), and/or AC3RL + siFBXO31 (20 nM) for 24 h. (**A**) SA-β-Gal staining in HAECs from each treatment group. (**B**) Intracellular reactive oxygen species (ROS) production in HAECs. (**C**) Immunofluorescence staining showing γH2AX foci in HAECs from each treatment group. Positively stained cells were quantified and are represented as a proportion of total cells. For the quantitative analyses, *n* = 3 per group. ** *p* < 0.01 vs. control; # *p* < 0.05, ## *p* < 0.01 vs. AC3RL.

**Figure 4 biomedicines-10-00854-f004:**
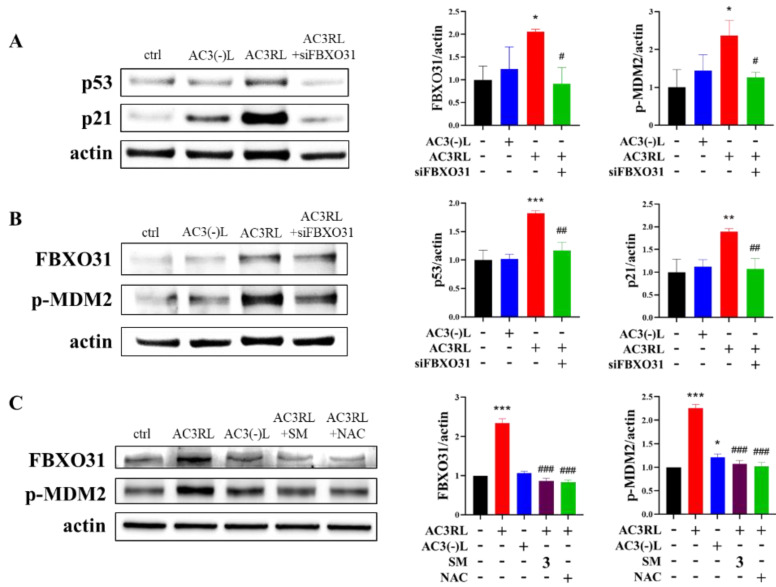
The signaling of AC3RL-induced senescence in HAECs. HAECs were incubated with PBS (control, ctrl), AC3(-)L (25 μg/mL), AC3RL (25 μg/mL), AC3RL + siFBXO31 (20 nM), AC3RL + NAC (5 mM), or AC3RL + SM (3 μM) for 24 h (**A**–**C**). Then, the protein expressions of p53 and p21 (**A**) or FBXO31 and p-MDM2 (**B**,**C**) were determined using Western blotting. Actin was used as an internal control. For the quantitative analyses, *n* = 3 per group. * *p* < 0.05, ** *p* < 0.01, *** *p* < 0.001 vs. control; # *p* < 0.05, ## *p* < 0.01, ### *p* < 0.001 vs. AC3RL.

**Figure 5 biomedicines-10-00854-f005:**
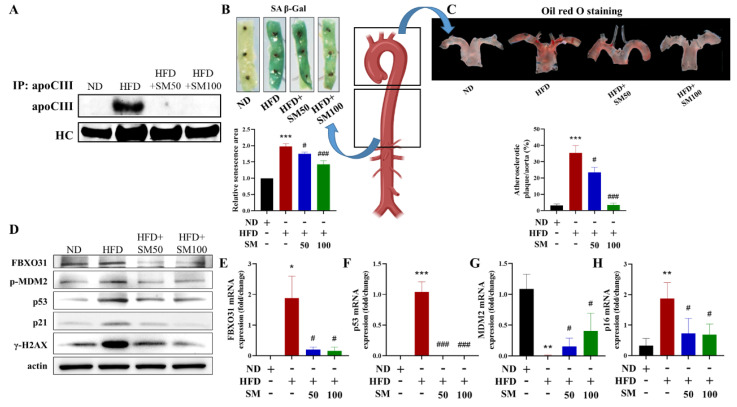
Pro-senescent effect of endogenous AC3RL in hamsters. Syrian hamsters were fed chow diet (control), HFD, or HFD supplemented with 50 mg/kg SM (HFD + SM50) or 100 mg/kg SM (HFD + SM100) via oral gavage for 18 weeks (*n* = 10 per group). (**A**) ApoC3 in LDL, (**B**) SA-β-gal staining, (**C**) Oil Red O staining, (**D**) Western blotting, (**E**) FBXO31 mRNA, (**F**) p53 mRNA, (**G**) MDM2 mRNA, (**H**) p16 mRNA. * *p* < 0.05, ** *p* < 0.01, *** *p* < 0.001 vs. ND group; # *p* < 0.05, ### *p* < 0.001 vs. HFD group.

**Figure 6 biomedicines-10-00854-f006:**
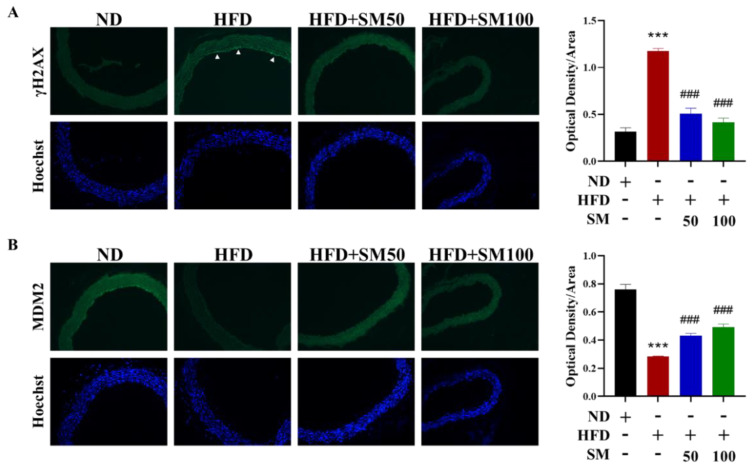
Immunofluorescence staining for γH2AX and MDM2. Syrian hamsters were fed chow diet (control), HFD, or HFD supplemented with 50 mg/kg SM (HFD + SM50) or 100 mg/kg SM (HFD + SM100) via oral gavage for 18 weeks (*n* = 10 per group). (**A**) Cross-sections of thoracic aorta tissues from treated mice were stained for γH2AX, a reliable marker of DNA double-strand breaks, using immunofluorescence. The endothelium was stained with positive staining as indicated by arrows. (**B**) Immunofluorescence staining of MDM2. Hoechst 33342 was used as a nuclear marker. Quantification of the percentage of cells with the presence of H2AX or MDM2. *** *p* < 0.001 vs. ND group; ### *p* < 0.001 vs. HFD group.

**Figure 7 biomedicines-10-00854-f007:**
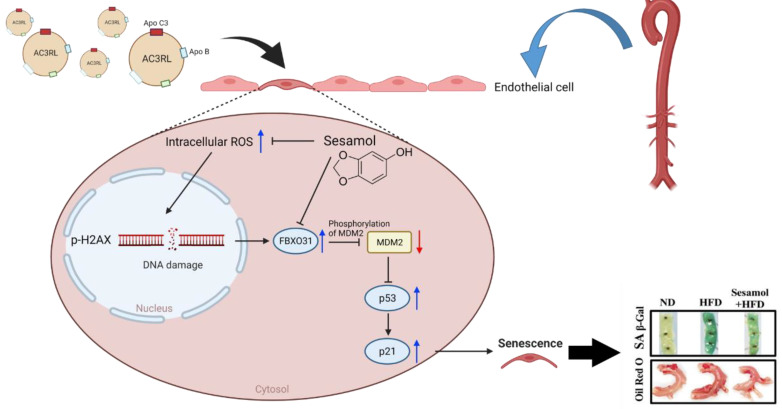
Scheme of AC3RL-induced cell senescence and how sesamol inhibited it.

**Table 1 biomedicines-10-00854-t001:** Real-time PCR primers of target genes.

Gene Name	Primer (5′→3′) Forward	Primer (5′→3′) Reverse
*GAPDH*	TTGTTGCCATCAATGACCCCTT	CGTTCTCAGCCTTGACTGTGCCTT
*FBXO31*	GTGGAGATCTTCGCCTCGCT	TCACAGACGCCATACTCCTCG
*MDM2*	CACAGGTCCCTTTCCTTTGA	TGAATCCTGATCCAGCCAAT
*p53*	CCCCCAAAGAGTGCTAAACGA	CAGTTCCAAGGCCTCATTCAA
*p16(INK4A)*	AGAGGTTCGGGCTTTGCT	CTACTTGGGTGTTGCCCATC

## Data Availability

An article containing the findings of this investigation is available online.

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
