# Peer review of "Apolipoprotein C3-Rich Low-Density Lipoprotein Induces Endothelial Cell Senescence via FBXO31 and Its Inhibition by Sesamol In Vitro and In Vivo"

_biomedicines, 2022, doi:10.3390/biomedicines10040854_

Round 1

Reviewer 1 Report

Review

The article is devoted to topical problems of modern medicine: aging, endothelial dysfunction. This is evidenced by the continuous growth of publications. There are no publicly available papers that fully disclose the role of FBXO31 in the aging of HAEC caused by Apolipoprotein C3. Therefore, this publication is relevant and timely. In addition, the article presents the results of an attempt to prevent Apolipoprotein C3-induced aging of HAEC Sesamol.

The authors have done a lot of work that has allowed us to substantiate the role of FBXO31 in the development of Apolipoprotein C3-induced HAEC aging. In addition, the authors at the experimental level have convincingly proved the possibility of correcting the caused pathological condition of Sesamol.

The study was carried out at a high technical level using the latest research methods. The reliability of the results obtained is achieved by an adequately constructed design of the experiment, the use of a sufficient number of experimental animals in each group, and adequate statistical processing of the scientific data obtained.

The article is well illustrated. The drawings are complete and informative. They allow unambiguously interpreting the received data.

The discussion is structured logically. The discussed provisions logically follow from the results obtained and the literature data provided.

Remarks

1. Bring the abbreviated or full form - sesamol or SM - to uniformity in figure captions.

2. The first sentence in the Discussion "In this study, we demonstrated, for the first time, that human AC3RL induces EC senescence both in vivo and in vitro" does not look good. Since human AC3RL was not injected into hamsters. This sentence needs to be redone.

It is recommended to print after minor revision.

Reviewer 2 Report

In their study the authors demonstrated that human AC3RL induces endothelial cell senescence both in vivo and in vitro. At least in vitro it was shown to be voia FBX031. The text reads fluently and I did not find main issues. However, minor modifications should be faced before considering this work for publication.

--> Introduction section

- lines 50-52: please quote this sentence. A reference is lacking.

- lines 66-67: which study did the authors refer to? It is not referenced.

--> Methods section

- Please provide references of antibodies used in all the assays described herein

- Line 137: please specify elution buffer composition

--> Results section

-  Although the contribution of FBX031 is elegantly addressed, the effects of sesamol on p21 and p53 directly in endothelial cells is not shown. Why?

--> Discussion section

- Lines from 255 on: the expression "HFD led to ApoC3 over- 255 expression on LDL" is not supported by data. ApoC3 is mainly expressed in the liver. The authoirs should refer to their abundance, which could be explained by other mechanisms, e.g. lipoprotein remodeling. Neither mRNA or protein analysis are shown.

- the hamster HFD group elevates ApoC3 in LDL compared with the control group, which is the underlying mechanism? It was not discussed.
